# Spillover of volatility among financial instruments: ASEAN-5 and GCC market study

**Nevi Danila** *

Finance Department, College of Business Administration, Prince Sultan University, Riyadh, Saudi Arabia

* ndanila@psu.edu.sa

**Data Availability Statement:** All relevant data are within the paper.

**Funding:** The authors would like to acknowledge the support of Prince Sultan University for paying the Article Processing Charges (APC) of this publication

## Abstract

The research examines a comovement and spillover of volatility among foreign exchange, conventional and shariah stock markets in Association of South East Asian Nation-5 (ASEAN) countries and Gulf Cooperation Council (GCC) countries. Generalized Autoregressive Conditional Heteroskedasticity—Baba, Engle, Kraft and Kroner (GARCH-BEKK) and Dynamic Conditional Correlation (GARCH-DCC) models are used to capture the correlation and transmission volatility of the markets. The overall results show that both the Shariah and the conventional stock indices respond similarly to each country's currency. A bidirectional (two-way relationship) volatility spillover exists only in Malaysia and a unidirectional (one-way relationship) volatility is observed in Indonesia, Singapore, Thailand, and Bahrain. The rest of the markets–the Philippines, Saudi Arabia, and the United Arab Emirates (UAE)–do not have any volatility spillover evidence. Based on DCC outcomes, the conventional and Shariah stock in ASEAN-5 countries and GCC countries reveal the market efficiency, i.e., a positive high conditional correlation. Only Bahrain shows less efficiency than the other countries. It implies no portfolio diversification advantage in conventional and Shariah stock indices. Contrarily, currency and stock (conventional and Shariah) markets provide portfolio diversification benefits for all ASEAN-5 and GCC countries.

## Introduction

International equity investments have been rapidly growing in the last three decades. It leads to an increase in the interdependency between stock returns and currency returns. In the meantime, the development of the Shariah stock market has also been growing swiftly. Hence, the interdependence among the three markets is highly possible. Studies also report that the Shariah stock market investors' behaviour is not significantly different from the conventional market. The increasing interdependency brings the consequences of volatility transmission among those markets. Numerous scholars investigated the dynamic connectedness between conventional and Shariah stocks [1–9].

The researchers also investigate a volatility spillover across two markets–conventional stock and foreign exchange market [10–14]. Additionally, it has been noticed that the Shariah stock market has played a significant role in the economy's growth. Nevertheless, to the best of our knowledge, the study on the transmission of volatility across the Shariah stock and foreign

**Competing interests:** The authors have declared
that no competing interests exist.

exchange market has not been done, mainly focusing on the Association of South East Asian Nations-5 (ASEAN) and Gulf Cooperation Council (GCC) markets. Therefore, developing financial theories and models can benefit from this study. It might improve existing models by combining insights from the interaction between foreign exchange, conventional and Sharia stock markets.

The additional justifications for why these studies are crucial are elaborated as follows. The study assists investors and portfolio managers in risk management and diversification by optimizing allocating funds across different asset classes. The spillover volatility between these markets can provide insight into the efficiency of information transmission, which affects investment decisions and market behaviour. The study can provide information about the macroeconomic environment. It may demonstrate the significance of global trade and economic issues in determining the performance of Sharia-compliant firms. The research might provide information to the policymakers for making decisions that promote sustainable growth and monetary and economic stability.

The structure of this study is outlined in the following. After investigating the shariah stock and foreign exchange market's volatility spillover by employing the GARCH-BEKK model in ASEAN-5 markets (Malaysia, Indonesia, Singapore, Thailand, and the Philippines) and three GCC markets (i.e., UAE, Saudi Arabia and Bahrain), we compare it to a spillover of conventional stock market and foreign exchange market. We expect a different outcome between the Shariah and conventional stock markets. Next, we explore the dynamics of the interaction of two-pair markets by employing the GARCH-DCC approach. Furthermore, we expect different results across the ASEAN-5 and GCC markets. As we noticed, an exchange rate fluctuation in GCC countries is relatively stable, i.e., the currency tends to pegs against the U.S. dollar. Finally, the hedge ratios and the optimal portfolio weights are estimated to convey the implication of portfolio diversification and currency risk management.

The remainder of the study proceeds as follows. The following section describes the theoretical background of the study. Then, we provide the data and details of the methodology used for the study in the data and methods section. The results and discussion section elaborates on the empirical findings and the recommendation based on data analysis. Finally, the last section presents the conclusion that indicates the relevance of our findings

## Comovement and spillover volatility in the financial market

Two models linked the stock prices and the currency is the flow-oriented and the stock-oriented model. The flow-oriented model is suggested by Dornbusch and Fischer [15]. The model suggests that stock rates and exchange prices have a positive relationship. The model comes from the concept that the country's trade balance or current account balance determines the exchange rate. It posits that fluctuation in the currency affects the nation's trade balances and competitiveness, which leads to changes in real income and input. For example, the local currency depreciation will drive the export price down; consequently, domestic companies will be more competitive. A high export will boost the domestic firms' stock price; eventually, the wealth of domestic firms will be propped up. Hence, the theory describes the causality as from the movement of the currency to the stock price.

The stock-oriented model is proposed by Branson and Henderson [16] and Frankel [17]. The model suggests that the supply and demand of financial assets, i.e., bonds and stock (portfolio), determines the exchange rate. Further, the Portfolio balance model argued that the stock price and foreign exchange have a negative correlation. Investors hold both domestic and foreign assets and domestic and foreign currency. When domestic assets' price increases, the inventors tend to buy more by selling foreign assets. It means that the demand for domestic

currency increases, and foreign currency supply is also increased. Consequently, the appreciation of the domestic currency is inevitable.

Many scholars have conducted an empirical study on the volatility transmission between the stock and foreign exchange markets. Jebran and Iqbal [12] examine volatility spillover effects between two markets in ASIAN countries, such as Japan, China, Pakistan, Hong Kong, Sri Lanka and India, employing exponential generalized autoregressive conditional heteroskedastic (EGARCH). They found that bidirectional leverage volatility spillover exists between the two markets in Sri Lanka, Pakistan, Hong Kong and China. In comparison, the Indian market has unidirectional volatility spillover. On the other hand, there is no evidence of volatility transmission in the Japanese market.

Moreover, a study reveals that all markets have asymmetry volatility, i.e., negative news will create higher volatility than positive information of similar magnitude. In the same context, Chkili [13] observed the spillover of volatility across two financial assets: currency and stock prices in developing countries, namely Argentina, Brazil, Mexico, Indonesia, Malaysia, Singapore, Hong Kong, and South Korea. The relationship is bidirectional; however, stock markets' movements affect the exchange rates more than the impact of currency movement towards a stock exchange price. The author suggested that the two markets are integrated with emerging markets. Furthermore, the optimal portfolio holding is also estimated, which results in the investors being advised to hold the currency more than the stock. By combining this proportion, the investors will minimize portfolio risk without sacrificing the return expectation.

Furthermore, studying the volatility transmission in Brazil, Russia, India, China, and South Africa (BRICS) countries, Chkili and Nguyen [18] reported that the BRICS countries' returns develop following two regimes: low regime volatility and high volatility. During both the calm and turmoil periods, stock markets influence more on exchange rate markets. Kang and Yoon [10] reported similar results; there is evidence of symmetry and unidirectional spillover of volatility from the Korean Stock market (KOSPI) to the Korean currency (KRW) after post-crisis, and there are spillover effects before the crisis. It implies that the crisis enhances linkages between two markets. However, Beer and Hebein [11] came up with the opposite result; there is evidence of unidirectional spillover from currency markets to stock exchange markets for the U.S., Canada, Japan, South Korea, and India. They argued that a depreciation of the exchange rate leads to a drop in the stock price as the currency's depreciation is followed by higher inflation in the future, which creates scepticism for investors on companies' future performance.

Using a different method–forecast-error variance decomposition framework of a VAR model–Grobys [19] investigated the volatility transmission between the U.S. and its three large trading partners (Canada, European Union, and Japan). The author argued that volatility spillover is at a high level before the economic turbulence occurs. During economic turmoil, the volatilities of financial markets are driven by a common factor. In normal conditions, the fluctuation of the financial market is driven by an individual factor, which is no evidence of volatility spillover effects. Regarding the economic events, Antonakakis [20] reported that comovements and bidirectional spillover between the U.S. dollar and four major internationally traded currencies are positively associated with both pre and post-Euro introduction. However, the volatility spillover and its comovement are lower in the post-euro period.

Regarding the transmission volatility between traditional and Shariah stock markets, many scholars reported evidence of dynamic comovements and both unidirectional and bidirectional volatility spillover. For example, Hasan [7] found several findings of the volatility spillover between Bangladesh's traditional and Islamic stock markets. First, conventional stock prices have a long-term bidirectional effect on Islamic stock prices. Second, both markets show evidence of clustering volatility and the leverage effects, i.e., the bad information gives

more reaction on the markets than good information. Third, GARCH-BEKK results show volatility transmission from a conventional to an Islamic market. Last, the GARCH-DCC model reveals the presence of conditional and time-varying connectedness between the two markets.

Nazlioglu [21], using the three U.S. conventional stock markets index (SPA500), Europe (SPEU) and Asia (SPAS50TR), and the Dow Jones Islamic stock (DJIM) index, reported the same findings during pre-, in, and post-2008 crisis. The authors suggested that volatility during turmoil and crisis periods will be longer-lasting than volatility during calm and normal periods. Additionally, there is evidence of mutual volatility transmission between the Islamic stock market and its counterparts. Finally, they imply that the Islamic stock market is not a safe haven during crises; in other words, the Sharia-based principles market is similar to its conventional market. On the other hand, Ahmad et al. [6] argued that on top of a directional interdependence across DJIMI and traditional markets, the DJIM exhibited a disassociate pattern during the 2008 financial crisis. The European interest rate baseline and prices of crude oil construct a Sharia-screened stock index as an effective hedging instrument in the financial crisis.

In the same context, Jebran et al. [5] reported a significant short and long-term relationship between traditional and Islamic indices in the Pakistan market. The asymmetric bidirectional volatility transmission between these markets also exists, suggesting domestic investors have diversification opportunities at a low level. International investors also benefit from diversification by adding the indices to their portfolios. Nevertheless, Kim and Sohn [3] concluded that there is a unidirectional volatility spillover from the traditional stock price index to the Islamic stock price index in the U.S. market. The empirical results also indicate no evidence of volatility transmission from the U.S. conventional stock index to the globally diversified DJIM index based on Sharia values.

Furthermore, the authors argued that the benefits of diversification among several Islamic stock indices are more appealing than those among their conventional counterparts. Another study by Majdoub et al. [4] reveals no correlation across the Indonesian and developed markets for both traditional and Islamic stock prices. Hence, investors can diversify their investments internationally to reduce risk. In contrast, there are high linkages across the developed markets for traditional and Islamic indices. Besides, the Islamic index is strongly connected with its traditional counterpart for each country.

## Data and methods

Our study utilizes daily data on the traditional and Islamic stock indexes as well as the exchange rate (against the U.S. dollar) for each of the countries represented by the ASEAN-5 markets–Indonesia, Singapore, Thailand, Philippines, and Malaysia, as well as the GCC markets–Saudi Arabia, Bahrain, and the United Arab Emirates (UAE). The indexes of these nations will be analyzed with those of the Islamic countries. The data comes from Bloomberg. The data scope span differs from one country to the next, depending on the accessibility of the data. All data are converted into rate of return using the formula $R_t = 100 * ln\left[\frac{P_t}{P_{t-1}}\right]$. Table 1 shows the details of our data.

We utilized a multivariate GARCH-BEKK approach to pinpoint the fluctuation and spillover among the sharia/conventional stock indices and currency rates in the mentioned countries. Below is the formula of the GARCH-BEKK

$$H_t = CC' + \sum_{i=1}^{k} A_i \varepsilon_{t-1} \varepsilon\prime_{t-1} A\prime_i + \sum_{i=1}^{k} G_i H_{t-1} G\prime_i, \qquad [1]$$

where *C, $A_i$, and Bi = N×N matrices and C is triangular*. All the positive-definite diagonal notation is guaranteed by Eq (1).

**Table 1. Data description\*.**

| Name | Period |
|------|--------|
| Indonesia Composite Index (JKSE) | Jan. 02 2015 to Oct. 01 2020 |
| Jakarta Islamic Index (JII) | Jan. 02 2015 to Oct. 01 2020 |
| Indonesia Rupiah (Rp) | Jan. 02 2015 to Oct. 01 2020 |
| MSCI Malaysia Index (MXMY) | Jan. 07 2015 to Oct. 04 2020 |
| FTSE Bursa Malaysia Hijrah Shariah Index (FBMHS) | Jan. 07 2015 to Oct. 04 2020 |
| Malaysian Ringgit (MYR) | Jan. 07 2015 to Oct. 04 2020 |
| MSC Singapore Islamic Index (MISG) | Jan. 04, 2015, to Oct. 03 2010 |
| Straits Time Index (STI) | Jan. 04, 2015, to Oct. 03 2010 |
| Singapore Dollar (SGD) | Jan. 04, 2015, to Oct. 03 2010 |
| Stock Exchange of Thailand Index (SET) | Jan. 09 2015 to Oct. 05 2020 |
| MSCI Thailand Islamic Index (MITH) | Jan. 09 2015 to Oct. 05 2020 |
| Thai Bath (TBH) | Jan. 09 2015 to Oct. 05 2020 |
| Philippines Stock Exchange Index (PCOMP) | Jan. 04 2015 to Oct. 02 2020 |
| MSCI Philippines Islamic Index (MIPH) | Jan. 04 2015 to Oct. 02 2020 |
| Philippines Peso (Peso) | Jan. 04 2015 to Oct. 02 2020 |
| Dubai Financial Market General Index (DFMGI) | Jan. 06 2015 to Oct. 05 2020 |
| S&P UAE Shariah Domestic T.R. (SPSHDADT) | Jan. 06 2015 to Oct. 05 2020 |
| UAE Dirham (Dirham) | Jan. 06 2015 to Oct. 05 2020 |
| Tadawul All Share Index (SASEIDX) | Jan. 07 2015 to Oct. 05 2020 |
| S&P Saudi Arabia Shariah Index (SPSHSA) | Jan. 07 2015 to Oct. 05 2020 |
| Saudi Riyal (SAR) | Jan. 07 2015 to Oct. 05 2020 |
| Bahrain Bourse All Share Index (BHSEASI) | Jan. 06 2015 to Oct. 05 2020 |
| S&P Bahrain Shariah Bahrain Dinar (SBHHPL) | Jan. 06 2015 to Oct. 05 2020 |
| Bahrain Dinar (BHD) | Jan. 06 2015 to Oct. 05 2020 |

\* All currency against USD. Saudi Arabia, UAE and Bahrain are among the most economically significant in the GCC. Saudi Arabia has the largest economy in the region, followed by the UAE, and Bahrain is known for its financial services sector. These countries play a crucial role in shaping the economic landscape of the GCC.

We also take into account the following basic GARCH (1,1):

$$H_t = CC' + A_1\varepsilon_{t-1}\varepsilon'_{t-1}A'_1 + G_1H_{t-1}G'_1. \tag{2}$$

Then, the BEKK formula is as below

$$H_t = CC' + \begin{bmatrix} a_{11} & a_{12} \\ a_{21} & a_{22} \end{bmatrix} \begin{bmatrix} \varepsilon^2_{1,t-1} & \varepsilon_{1,t-1}\varepsilon_{2,t-1} \\ \varepsilon_{2,t-1}\varepsilon_{1,t-1} & \varepsilon^2_{2,t-1} \end{bmatrix} \begin{bmatrix} a_{11} & a_{12} \\ a_{21} & a_{22} \end{bmatrix}' + \begin{bmatrix} g_{11} & g_{12} \\ g_{21} & g_{22} \end{bmatrix} \begin{bmatrix} h_{11,t-1} & h_{12,t-1} \\ h_{21,t-1} & h_{22,t-1} \end{bmatrix}$$
$$\times \begin{bmatrix} g_{11} & g_{12} \\ g_{21} & g_{22} \end{bmatrix}', \tag{3}$$

Where ARCH effects are captured by matrix *A* and GARCH effects are reflected by matrix *G*. The diagonal parameters determine how their conditional variances are impacted by their historical shocks and volatility. The off-diagonal factors determine the volatility spillover from each market.

Each conditional variance in the BEKK model is disintegrated into its GARCH and ARCH components.

$$h_{11,t} = a^2_{11}\varepsilon^2_{1,t-1} + a^2_{21}\varepsilon^2_{2,t-1} + 2a_{11}a_{21}\varepsilon_{1,t-1}\varepsilon_{2,t-1}, \tag{4}$$

The ARCH fluctuation in shariah/conventional stocks indices depends on the squares and the cross-products of the previous fluctuation related to shariah/conventional stock indices and exchange rate. Then, $a_{11}$ and $a_{21}$ catch the results of past squared fluctuation in each market on today's volatility in the shariah/traditional stocks.

The GARCH variables of the shariah/traditional stocks conditional variances are as below

$$h_{11,t} = g_{11}^2 h_{11,t-1} + g_{21}^2 h_{22,t-1} + 2g_{11}g_{21}h_{12,t-1}, \qquad [5]$$

Then, $g_{11}$ and $g_{21}$ capture the impacts of past fluctuation in each market on today's volatility in the shariah/conventional stocks.

Furthermore, our study employs the GARCH-DCC approach suggested by Engle [22] to examine the fluctuation of conditional relationships among the markets. The formula is below.

$$H_t = D_t R_t D_t \qquad [6]$$

Where $D_t$ is a $N$ x $N$ diagonal matrix of time-varying standard deviations from univariate GARCH models. $R_t$ is a correlation matrix defined as

$$R_t = diag\left(q_{11,t}^{-\frac{1}{2}} \ldots q_{22,t}^{-\frac{1}{2}}\right) Q_t diag\left(q_{11,t}^{-\frac{1}{2}} \ldots q_{22,t}^{-\frac{1}{2}}\right), \qquad [7]$$

where the 2 x 2 symmetric positive definite matrix, and $Q_t = (q_{ij,t})$ is stated as

$$Q_t = (1 - \alpha - \beta)\bar{Q} + \alpha\varepsilon_{t-1}\varepsilon'_{t-1} + \beta Q_{t-1}, \qquad [8]$$

where standardized residuals $\varepsilon_t = \varepsilon_{it}/\sqrt{h_{iit}}$ and $\alpha$ and $\beta$ are non-negative scalar parameters with the $\alpha+\beta<1$ constraint. The value of $\alpha+\beta$ close to 1 indicates the existence of a high continuation in the conditional variance and mean returning [23]. The univariate GARCH (1,1) approach is used to get the standardized residuals for assessing the connectedness variables of the DCC approach.

## Results and discussion

### Descriptive statistics

Tables 2–4 report the descriptive statistics of the conventional stock index, shariah stock index, and currency returns consecutively. The series represents daily returns for Indonesia,

**Table 2. Descriptive statistics of conventional stock index return.**

| Variable | Indonesia | Malaysia | Singapore | Thailand | Philippines | Saudi Arabia | UAE | Bahrain |
|---|---|---|---|---|---|---|---|---|
| Mean | -0.0038 | -0.019623 | -0.013569 | -0.007623 | -0.005407 | -0.034468 | -0.066937 | -0.006557 |
| SD | 1.0550 | 2.915404 | 4.166179 | 4.333698 | 5.040766 | 5.829833 | 6.298723 | 4.151142 |
| Skewness | -0.1760 | 0.060726 | 0.448658 | -0.389823 | -0.051810 | -0.126180 | -0.688890 | -0.687749 |
| Kurtosis | 9.4055 | 8.079236 | 11.727487 | 13.033359 | 11.225959 | 7.917934 | 14.723746 | 11.945464 |
| Normality test (p-value) | 5145.54 (0.000)*** | 3830.27 (0.000)*** | 8340.69 (0.000)*** | 9674.52 (0.000)*** | 7084.09 (0.000)*** | 2821.46 (0.000)*** | 9922.92 (0.000)*** | 6446.13 (0.000)*** |
| ADF (p-value) | -35.3178** | -45.2347** | -44.7528** | -47.7801** | -43.4603** | -39.2450** | -41.7297** | -38.4709** |
| LM test (5 lags) | 0.7756 | 0.0001*** | 0.1905 | 0.0012*** | 0.03692 | 0.2267 | 0.0682* | 0.0260** |
| LM ARCH test (5 lags) | 0.0000*** | 0.0000*** | 0.0004*** | 0.0000*** | 0.0000*** | 0.0000*** | 0.0000*** | 0.0000*** |

***1% level of significance

**5% level of significance and

*10% significance level.

**Table 3. Descriptive statistics of shariah index return.**

| Variable | Indonesia | Malaysia | Singapore | Thailand | Philippines | Saudi Arabia | UAE | Bahrain |
|---|---|---|---|---|---|---|---|---|
| Mean | -0.019176 | -0.013306 | -0.023355 | 0.000716 | -0.020556 | -0.029491 | -0.008084 | -0.047008 |
| SD | 1.370732 | 3.233135 | 4.780184 | 6.741789 | 5.638182 | 5.674518 | 4.631609 | 9.140763 |
| Skewness | -0.037078 | 0.103837 | 1.692046 | -0.698330 | 0.081914 | -0.183029 | -1.152407 | -0.365330 |
| Kurtosis | 7.763098 | 15.161457 | 22.427772 | 11.012804 | 13.296024 | 8.431752 | 12.569790 | 7.183082 |
| Normality test (p-value) | 3495.72 (0.000) *** | 13507.38 (0.000) *** | 31017.47 (0.000) *** | 6993.44 (0.000) *** | 9938.25 (0.000) *** | 3202.31 (0.000) *** | 7410.27 (0.000) *** | 2324.15 (0.000) *** |
| ADF (p-value) | -37.2699** | -40.9875** | -42.8970** | -48.4984** | -43.6293** | -39.3060** | -41.3681** | -41.8722** |
| L.M. test (5 lags-p-value) | 0.2635 | 0.0058*** | 0.0087*** | 0.0020*** | 0.0289** | 0.2038 | 0.0746* | 0.0887* |
| LM ARCH test (5 lags) | 0.0000*** | 0.0000*** | 0.0000** | 0.0000*** | 0.0000*** | 0.0000*** | 0.0000*** | 0.0000*** |

***1% level of significance

**5% level of significance and

*10% significance level.

Malaysia, Singapore, Thailand, Philippines, Saudi Arabia, UAE, and Bahrain. The mean values for conventional and Shariah stock indices series are negative and near zero; only the Thai Shariah stock index has a positive mean close to zero. The mean currencies series are mixed; some have a positive value close to zero, while others have a negative value near zero. The standard deviations of conventional and Shariah stock indices returns are 1 to 3 for Indonesia and Malaysia. For the rest of the countries, the standard deviation is above 3.

Nevertheless, the currency series has a deviation below 1 for all countries. A small amount of variation can be seen in the currency series. According to this result, the price range of the currency return is constrained. The currency is hence at low-price volatility. As its skewness shows, the series does not follow the standard distribution curve. The sig-value of the normality test outcomes in the conclusion that the distribution is not normal for all series. Regarding skewness, most of the series is relatively symmetrical. This outcome implies that more indices returns are distributed around the mean value.

**Table 4. Descriptive statistics of currency return.**

| Variable | Indonesia | Malaysia | Singapore | Thailand | Philippines | Saudi Arabia | UAE | Bahrain |
|---|---|---|---|---|---|---|---|---|
| Mean | 0.012241 | 0.010991 | 0.001338 | -0.003391 | 0.005484 | -0.000059 | -0.000003 | 0.000022 |
| SD | 0.440186 | 0.429674 | 0.328546 | 0.298630 | 0.271343 | 0.020500 | 0.001520 | 0.056386 |
| Skewness | 0.254339 | -0.565742 | -0.178364 | 0.068736 | 0.221717 | -0.280666 | -0.635726 | 1.162135 |
| Kurtosis | 18.150044 | 6.657273 | 2.201828 | 1.514422 | 0.820584 | 36.065991 | 19.864738 | 143.615944 |
| Normality test (p-value) | 18833.13 (0.000) *** | 2678.97 (0.000) *** | 299.96 (0.000) *** | 131.22(0.000) *** | 48.90 (0.000) *** | 58493.97 (0.000) *** | 17978.68 (0.000) *** | 919796.14 (0.000) *** |
| ADF (p-value) | -32.6394** | -35.0845** | -38.3880** | -33.9640** | -38.0511** | -34.1366** | -43.393** | -2.5684 |
| LM test (5 lags-p-value) | 0.07263* | 0.0027*** | 0.0578** | 0.9760 | 0.3471 | 0.0000*** | 0.0017*** | 0.0000*** |
| LM ARCH test (5 lags) | 0.0000*** | 0.0000*** | 0.0000*** | 0.0000*** | 0.0000*** | 0.0000*** | 0.0000*** | 0.0000*** |

***1% level of significance

**5% significance level, and

*10% significance level.

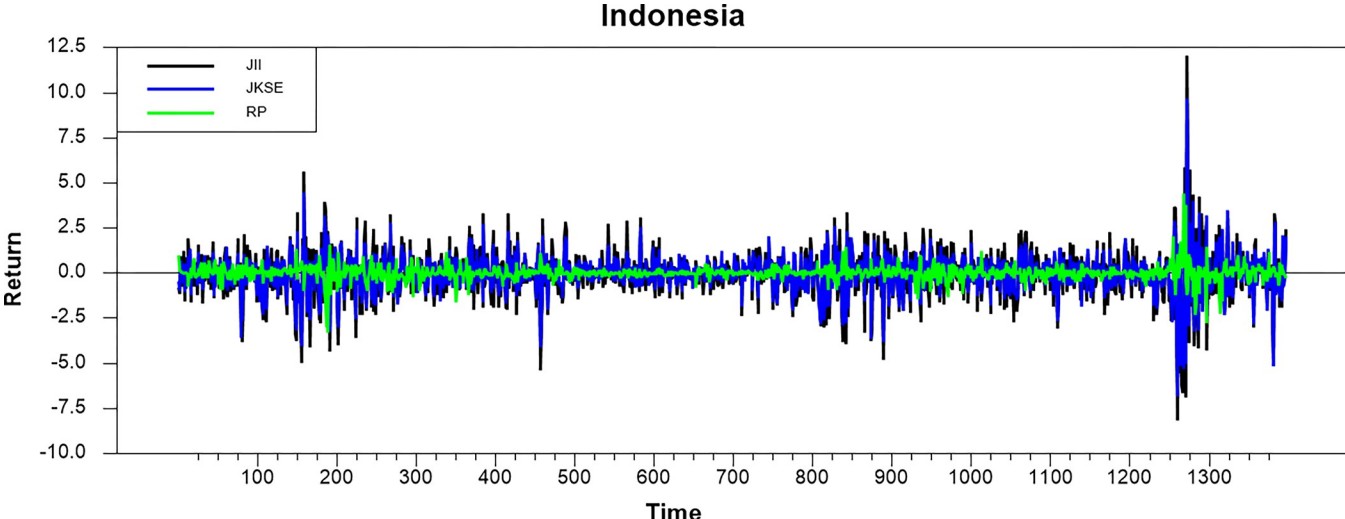

**Fig 1. Conventional and shariah stock index; and currency return volatility.**

Kurtosis indicates that the data have significant outliers, i.e., leptokurtic. Leptokurtic distribution indicates that an investor will undergo more significant variations (e.g., above three standard deviations from the mean), resulting in higher risk in returns. Regarding unit root data, the ADF test indicates stationary of the series. Finally, the null hypothesis of conditional homoscedasticity is rejected for all series at a 1% significance level; in other words, all series show ARCH effects—the evidence of clustering volatility.

Figs 1–8 show the movements of all returns for all countries. The graphs demonstrate that conventional and Shariah stock index returns move closer in the same direction for all countries, even though we spot that volatility spillover between both instruments does not have the same magnitude in several countries, such as the Thai, UAE, and Bahrain markets. The currency markets reveal much less volatility compared to the other two indices. Only the Indonesian currency market shows the highest volatility. Meanwhile, the GCC countries exhibit the least volatility.

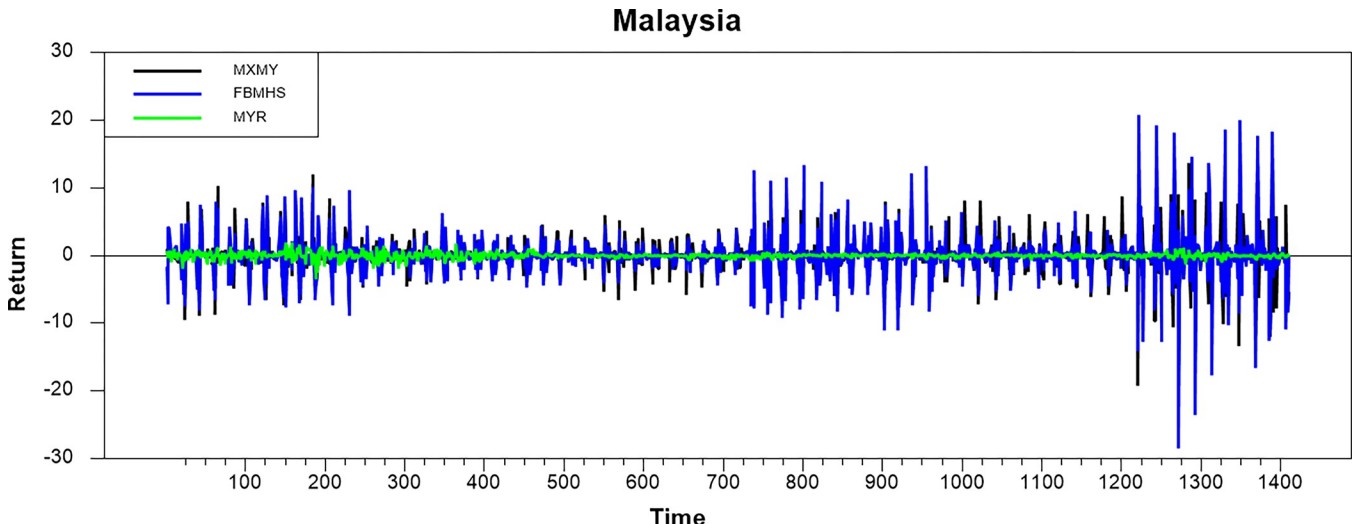

**Fig 2. Conventional and shariah stock index; and currency return volatility.**

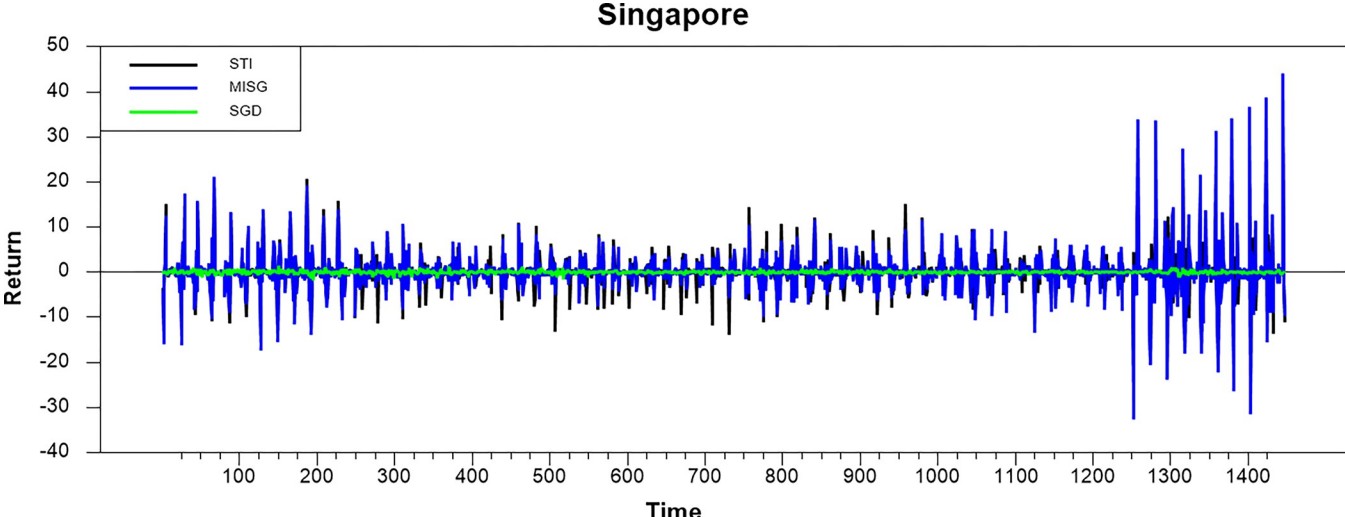

**Fig 3. Conventional and shariah stock index; and currency return volatility.**

## Volatility spillover

This part presents volatility spillover findings based on the GARCH-BEK method. Table 5 reports the estimates of BEKK parameters for all countries. "A" refers to ARCH parameters associated with particular markets. ARCH impact implies that shocks' effects are more pronounced in the subsequent period (high short-run volatility). "B" refers to GARCH parameters associated with specific markets. GARCH impact implies that the effects of shocks are more persistent (high long-run volatility).

### Foreign exchange market vs. conventional stock market

The series shows that the conditional variance of the currency and conventional stock markets depends on their past volatility; only the Philippines' conventional stock market does not

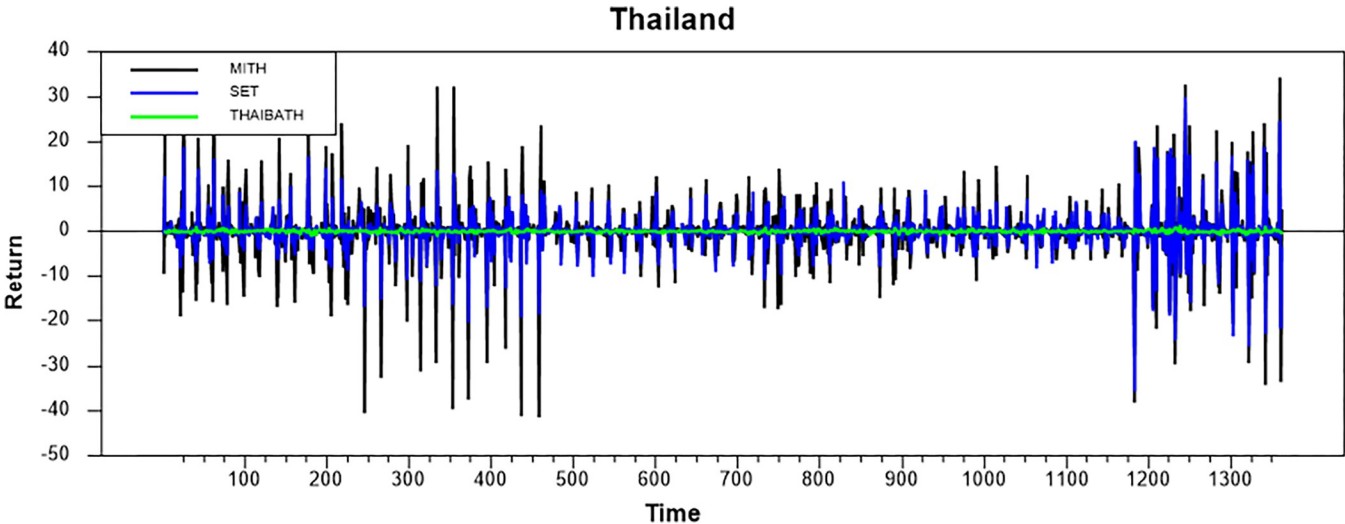

**Fig 4. Conventional and shariah stock index; and currency return volatility.**

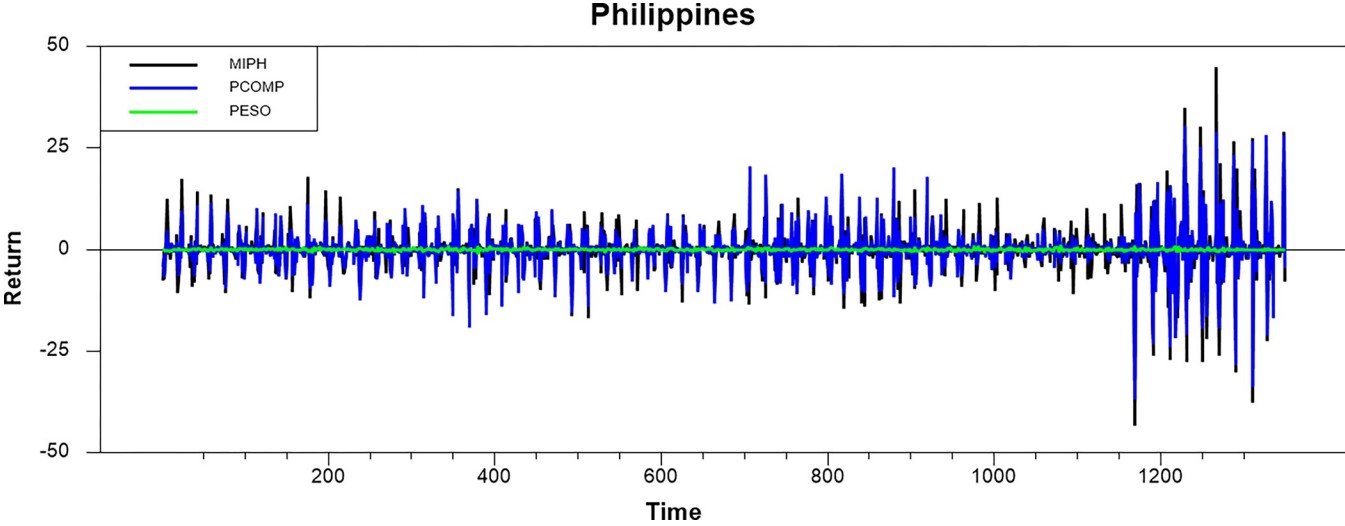

**Fig 5. Conventional and shariah stock index; and currency return volatility.**

depend on its past volatility. It means that the financial market's volatilities are induced by individual factors, as suggested by Grobys [19]. Estimated ARCH (high short-run volatility) demonstrates the significant effect of unidirectional transmission from the currency market to the conventional stock market for ASEAN-5 countries, except the Philippines; the negative coefficient of Indonesia and Malaysia markets means that the variance is affected more when the shocks move in opposite directions than they move in the same direction. The finding is supported by Beer and Hebein [11], who suggested that a decrease in stock price follows the currency's depreciation through the inflation channel, which leads to investors' pessimism about companies' future earnings. Meanwhile, only Bahrain, among the three GCC countries, exhibits a unidirectional spillover from the conventional stock market to the currency market, supported by Kang and Yoon [10]. The authors reported that the volatility transmission is from the Korean stock market to the Korean foreign exchange market.

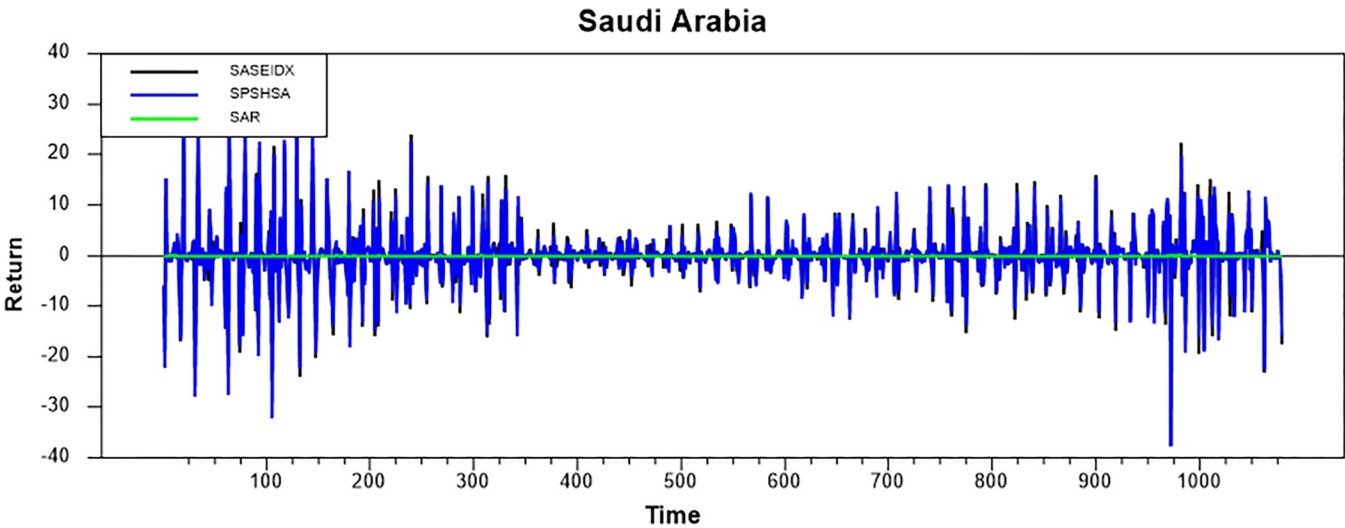

**Fig 6. Conventional and shariah stock index; and currency return volatility.**

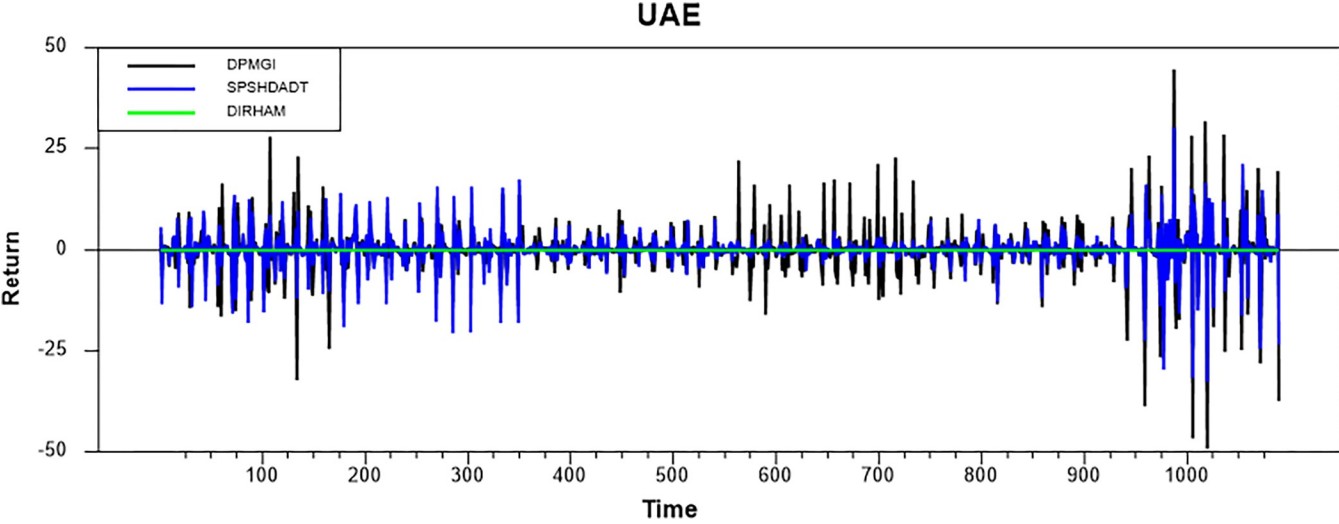

**Fig 7. Conventional and shariah stock index; and currency return volatility.**

GARCH (high long-run effect of volatility) parameters reported that the spillover is only revealed in the Malaysia and Singapore markets. The spillover is bidirectional in Malaysia, unidirectional from the stock market to Singapore's currency market. The unidirectional transmission findings are not consistent with the existing studies, such as Chkili [13]; Jebran and Iqbal [12] argue that the spillover is bidirectional and asymmetry between both conventional stock and currency markets, i.e., there is a two-way relationship of volatilities and the effect of currency movement towards stock index is higher than the impact of the stock index on currency

## Foreign exchange market vs. shariah stock market

The results suggested that the Shariah stock market's conditional variances depend on its own past volatility for all countries. A unidirectional spillover from the currency market to the

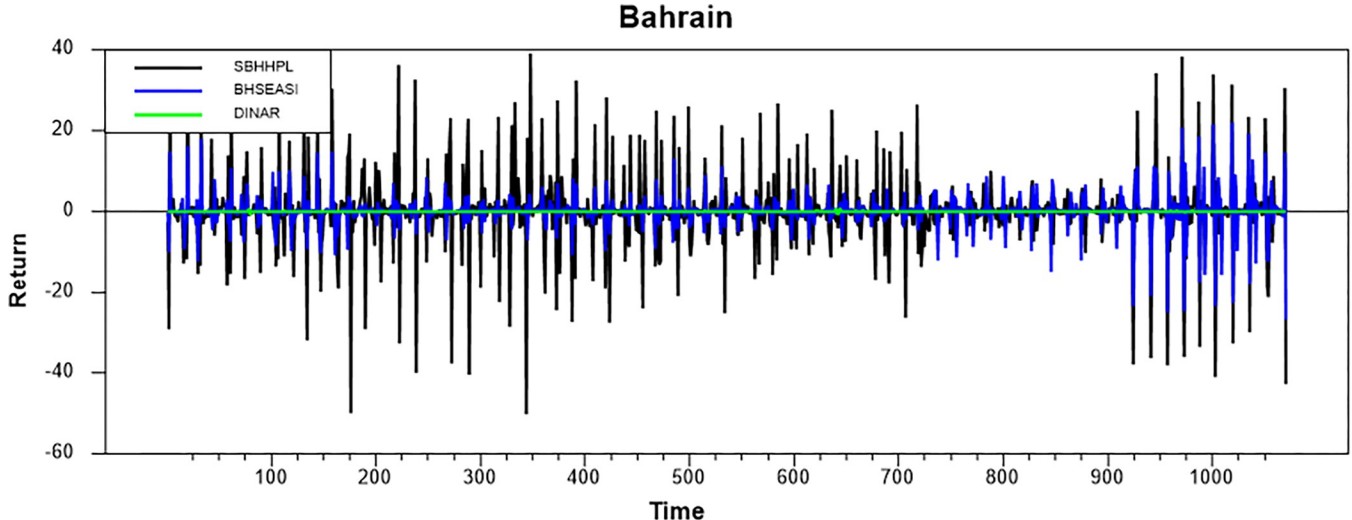

**Fig 8. Conventional and shariah stock index; and currency return volatility.**

**Table 5. GARCH-BEKK currency, conventional stock index, and shariah stock index.**

MV-GARCH and BEKK: Estimation by BFGS/BHHH With Heteroscedasticity/Misspecification Adjusted Standard Errors[##] *1 = Currency, 2 = Conventional Stock Index, 3 = Shariah Stock Index*

| Variable | Indonesia Coefficient (p-value) | Malaysia Coefficient (p-value) | Singapore Coefficient (p-value) | Thailand Coefficient (p-value) | Philippines Coefficient (p-value) | Saudi Arabia Coefficient (p-value) | UAE Coefficient (p-value) | Bahrain Coefficient (p-value) |
|---|---|---|---|---|---|---|---|---|
| $C(1,1)$ | 0.04039 (0.000)*** | -0.0213 (0.007)*** | -0.0216 (0.044)** | 0.0350 (0.263) | 0.0366 (0.096)* | -0.0012 (0.234) | 0.0004 (0.129)*** | -0.000 (0.832)*** |
| $C(2,1)$ | 0.01063 (0.890) | -0.0361 (0.00)*** | 2.3424 (0.034)** | -2.3430 (0.000)*** | -1.4139 (0.380) | -1.3497 (0.008)*** | -0.0701 (0.934) | -2.4034 (0.350) |
| $C(2,2)$ | 0.1647 (0.0001)*** | 0.9405 (0.000)*** | 1.8263 (0.106) | 1.6637 (0.000) | 2.6950 (0.021)** | 3.885 (0.000)*** | 4.0826 (0.000) | 0.9423 (0.876) |
| $C(3,1)$ | -0.0347 (0.7520) | -0.0283 (0.000)*** | 1.7489 (0.252) | -2.0505 (0.016)** | 0.5180 (0.756) | -1.1742 (0.000)*** | 0.3789 (0.731) | -4.297 (0.729) |
| $C(3,2)$ | 0.2230 (0.000)*** | 1.0037 (0.000)*** | 2.3428 (0.010)*** | 4.284 (0.000)*** | 2.6422 (0.000)*** | 3.7053 (0.000)*** | 1.7620 (0.000)*** | -3.2827 (0.313) |
| $C(3,3)$ | -0.0000 (0.9999) | -0.0000 (0.989) | -0.0005 (0.999) | 0.6095 (0.777) | 0.3061 (0.774) | -0.5053 (0.000)*** | 1.7768 (0.000) | 5.0961 (0.571) |
| $A(1,1)$ | 0.3784 (0.0000)*** | 0.2329 (0.000)*** | 0.1525 (0.000)*** | 0.0621 (0.512) | -0.1915 (0.000)*** | 0.7560 (0.000)*** | 0.5626 (0.000)*** | 2.6187 (0.000)*** |
| $A(1,2)$ | -0.2063 (0.0275)** | -0.597(0.020)** | 0.8922 (0.093)* | 4.7703 (0.000)*** | 0.2703 (0.823) | 12.0938 (0.421) | 78.3137 (0.319) | 3.8198 (0.595) |
| $A(1,3)$ | -0.3460 (0.0091)*** | -0.623 (0.022)** | 0.4097 (0.549) | 5.7166 (0.000)*** | 0.2738 (0.838) | 10.9587 (0.388) | -23.5446 (0.721) | 8.8943 (0.436) |
| $A(2,1)$ | 0.0309 (0.4456) | -0.0021 (0.518) | -0.0072 (0.466) | -0.0051 (0.427) | 0.0004 (0.923) | 0.0004 (0.590) | -0.0000 (0.396) | 0.000 (0.016)** |
| $A(2,2)$ | 0.3493 (0.0000)*** | 0.4519 (0.000)*** | 0.1853 (0.196) | 0.5336 (0.000)*** | 0.1882 (0.611) | -0.2666 (0.025)** | 1.0321 (0.0158)** | 0.4386 (0.000)*** |
| $A(2,3)$ | 0.3309 (0.0059)*** | 0.6307 (0.000)*** | -0.2714 (0.102) | -0.1073 (0.647) | -0.3839 (0.463) | -0.5061 (0.000)*** | 0.2609 (0.154) | 0.1078 (0.687) |
| $A(3,1)$ | -0.0041 (0.9053) | -0.0065 (0.086)* | 0.0001 (0.983) | -0.0032 (0.342) | 0.0008 (0.810) | -0.0003 (0.758) | 0.0000 (0.251) | -0.000 (0.589) |
| $A(3,2)$ | -0.0902 (0.1410) | 0.7701 (0.000)*** | 0.5266 (0.000)*** | 0.0758 (0.466) | 0.2465 (0.415) | 1.1485 (0.000)*** | -0.3428 (0.300) | -0.0017 (0.966) |
| $A(3,3)$ | -0.0739 (0.3827) | 0.7231 (0.000)*** | 0.7698 (0.000)*** | 0.7025 (0.000)*** | 0.8296 (0.089)* | 1.376 (0.000)*** | 0.3588 (0.074)* | 0.483 (0.001)*** |
| $B(1,1)$ | 0.9270 (0.0000)*** | 0.9715 (0.000)*** | 0.9826 (0.000)*** | 0.9615 (0.000)*** | 0.9686 (0.000)*** | 0.7684 (0.000)*** | 0.7972 (0.000)*** | 0.3957 (0.000)*** |
| $B(1,2)$ | 0.0416 (0.2240) | -0.0409 (0.000)*** | 0.0667 (0.873) | -0.4469 (0.593) | 0.7181 (0.355) | -15.8835 (0.306) | 166.464 (0.254) | -0.0686 (0.949) |
| $B(1,3)$ | 0.0923 (0.0719)* | -0.0533 (0.000)*** | 0.0294 (0.916) | -0.3395 (0.706) | -0.3051 (0.734) | -15.569 (0.275) | 178.1755 (0.203) | 3.6547 (0.164) |
| $B(2,1)$ | -0.0234 (0.1014) | 0.0029 (0.022)** | 0.0149 (0.001)*** | 0.0148 (0.276) | -0.0071 (0.538) | -0.0003 (0.880) | 0.0000 (0.161) | 0.0009 (0.274) |
| $B(2,2)$ | 1.0731 (0.0000)*** | 0.7235 (0.000)*** | 0.5112 (0.006)*** | 0.2016 (0.532) | 0.828 (0.008)*** | -0.1216 (0.200) | 0.1574 (0.485) | 0.6283 (0.000)*** |
| $B(2,3)$ | 0.1439 (0.0542)** | -0.2981 (0.000)*** | 0.0568 (0.698) | -0.3496 (0.542) | 0.6928 (0.098)* | 0.5147 (0.000)*** | -0.1505 (0.180) | 0.6629 (0.360) |
| $B(3,1)$ | 0.0132 (0.1854) | 0.0034 (0.025)** | -0.0085 (0.009)*** | 0.0028 (0.735) | 0.0057 (0.523) | -0.0001 (0.943) | -0.0000 (0.215) | -0.001 (0.012)** |
| $B(3,2)$ | -0.0994 (0.0059)*** | -0.2599 (0.000)*** | -0.0792 (0.630) | 0.1132 (0.609) | -0.1591 (0.481) | 0.4231 (0.000)*** | 0.3907 (0.100) | 0.0409 (0.437) |
| $B(3,3)$ | 0.8623 (0.0000)*** | 0.7062 (0.000)*** | 0.6117 (0.000)*** | 0.5032 (0.204) | 0.1232 (0.698) | -0.223 (0.029)** | 0.7766 (0.000)*** | 0.1415 (0.675) |

*(Continued)*

**Table 5.** (Continued)

| | | | | | | | | |
|---|---|---|---|---|---|---|---|---|
| Log-Likelihood | -2850.0485 | -6031.6204 | -7449.9194 | -7410.6665 | -7257.7501 | -1114.8119 | -184.7819 | -4009.9942 |

\*\*\* denotes significance at the 1% level.

\*\* denotes significance at the 5% level.

\* denotes significance at the 10% level.

##HAC (Heteroscedasticity and autocorrelation consistent) adjusted standard errors are employed to overcome misspecification of the model.

Shariah stock market is found in the Indonesian and Thai markets, supported by Beer and Hebein [11]. The reason for the persistent effect spillover in the Indonesian market is that the Shariah index consists of the 30 most liquid stocks, leading to the fast response volatility of the market. Moreover, the information converted from the currency market is faster and more efficient than the Shariah stock market because of the matured currency market, as Maghyereh and Awartani suggested [24]. However, the findings are different from the existing literature, which states that a bidirectional spillover exists in the stock and currency market [12, 13, 18].

Meanwhile, the opposite unidirectional volatility transmission is reported in the Singapore and Bahrain markets. The Malaysian market is the only market with a bidirectional volatility spillover. On the contrary, there is no evidence of spillover in the Philippine, Saudi Arabia, and UAE markets. Thus, investors and fund managers can use these financial instruments to employ risk management strategies such as hedging or asset allocation adjustments. The overall results are similar to the conventional stock markets. It implies that both Shariah and the conventional stock market respond similarly to each country's foreign exchange markets.

## Conventional stock market vs. shariah stock market

Surprisingly, the volatility spillover is not observed in the Thai, Philippine, UAE, and Bahrain markets. It implies that those countries' financial markets are unaffected by common factors, i.e., individual factors influence the price movement. For example, in the Thai market, the top 10 constituents of the MCSI Thailand Islamic Index are not included in the top performers of the Thailand SET index. We argue that the outcomes differ from the existing studies, which report high volatility transmission between these two financial markets. In contrast, the bidirectional volatility transmission is reported in Indonesia, Malaysia, and Saudi Arabia markets.

**Table 6.  GARCH-DCC conventional stock index, shariah stock index, and currency.**

| MV-GARCH and DCC: Estimation by BFGS/BHHH With Heteroscedasticity/Misspecification Adjusted Standard Errors## | | | | | | | | |
|---|---|---|---|---|---|---|---|---|
| Variable | Indonesia Coefficient (p-value) | Malaysia Coefficient (p-value) | Singapore Coefficient (p-value) | Thailand Coefficient (p-value) | Philippines Coefficient (p-value) | Saudi Arabia Coefficient (p-value) | UAE Coefficient (p-value) | Bahrain Coefficient (p-value) |
| DCC ($A$) | 0.0285 (0.0134)\*\* | 0.0425 (0.0020)\*\*\* | 0.0094 (0.0000)\*\*\* | 0.0919 (0.000)\*\*\* | 0.1108 (0.000)\*\*\* | 0.2205 (0.000)\*\*\* | 0.0733 (0.000)\*\*\* | 0.3831 (0.000)\*\*\* |
| DCC ($B$) | 0.9333 (0.0000)\*\*\* | 0.9276 (0.0000)\*\*\* | 0.9879 (0.0000)\*\*\* | 0.7184 (0.000)\*\*\* | 0.5619 (0.000)\*\*\* | 0.5115 (0.000)\*\*\* | 0.8053 (0.000)\*\*\* | 0.2080 (0.000)\*\*\* |
| Log-Likelihood | -2863.8957 | -4998.0802 | -6081.8069 | -6181.0538 | -6421.7951 | 105.9376 | 1280.0709 | -1864.2746 |

\*\*\* denotes significance at the 1% level.

\*\* denotes significance at the 5% level.

\* denotes significance at the 10% level.

##HAC (Heteroscedasticity and autocorrelation consistent) adjusted standard errors are employed to overcome misspecification of the model.

In Indonesia's case, the finding is as expected since the Jakarta Islamic Index (JII) comprises the 30 most liquid Islamic shares included in the Indonesia Composite Index. Besides, Singapore shows a unidirectional spillover. The finding is supported by Hasan [7], Majdoub et al. [4] Nazlioglu et al. [21]. The Islamic index is strongly connected with its traditional counterpart.

### Comovement of the foreign exchange, conventional, and shariah stock markets

According to Table 6, all markets contain a small digit of DCC (A) and a significant digit of DCC (B) for specific markets, with a total value of about 1. The discovery demonstrates a conditional dynamic relationship. The DCC (B) element is essential and indicates that volatility has a high level of persistence. All series' DCC (A) and DCC (B) sums are smaller than 1, suggesting mean-reverting of the conditional correlation process. As a result, the relationship recovers to the long-term unconditional level following the crisis episodes.

Figs 9–16 plot the dynamic conditional relationship among the foreign exchange, traditional stock, and shariah stock for all countries. These figures spot high positive comovement between traditional and Shariah stock indexes for all countries. The figures reveal that the connectedness between conventional and Shariah is projected to vary between 0.75–1.00 most of the period, except the UAE market, which ranges from 0.5–0.95. Only the Bahrain market exhibits the broadest range for -.025–0.95, but most of the time, the range is between 0.25–0.75.

In the same context, we observe that each conventional and Shariah stock market's comovement shows the same behaviour towards the currency market for all countries. Due to all the stocks in the Shariah index, they are also included in the conventional stock index, even though some of the stocks are not constituted as the best performers.

Looking into the level of correlation, only the Indonesian market shows the highest negative comovement, which ranges between -.25 - -.75. It is consistent with the stock-oriented model

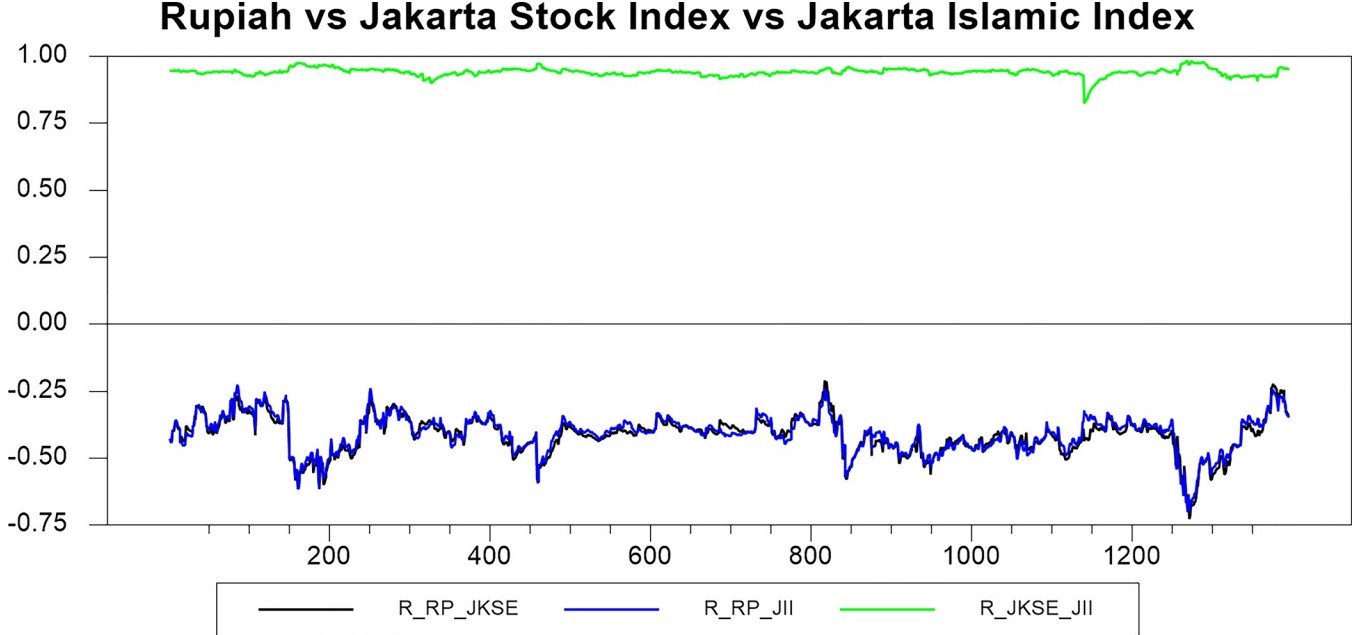

**Fig 9. Conditional correlation.**

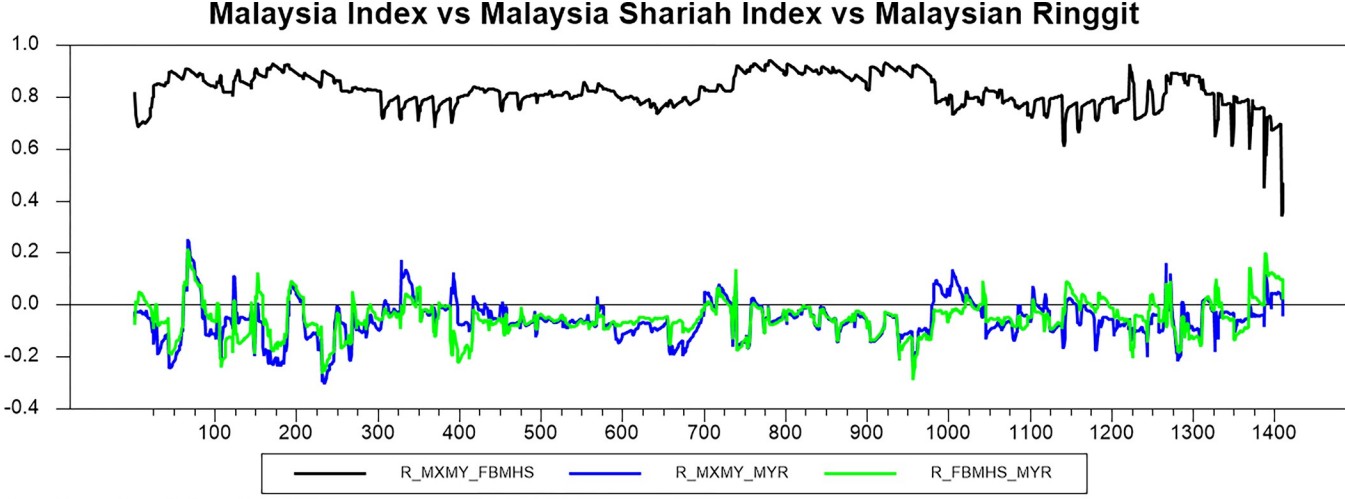

**Fig 10. Conditional correlation.**

theory that an increase in stock price will be followed by appreciating the home currency. For the rest of ASEAN-5 markets, the comovement range is between -.2–0 for most of the period. Meanwhile, Bahrain and UAE markets disclose that connectedness fluctuates around 0. Further, the Saudi Arabia market's comovement is not much different from the other GCC markets, but the comovement is slightly more volatile.

Based on the DCC outcomes above, we derive that the traditional and Shariah stock in ASEAN-5 countries and GCC countries markets demonstrate the markets' efficiency–only Bahrain shows less efficiency–which does not provide investors with any diversification advantages for including the assets in their portfolios. Nevertheless, currency and stock (conventional and Shariah) markets provide the advantage of portfolio diversification for all ASEAN-5 and GCC countries. Concerning portfolio diversification strategies, investors need to increase their awareness of market comovements accordingly.

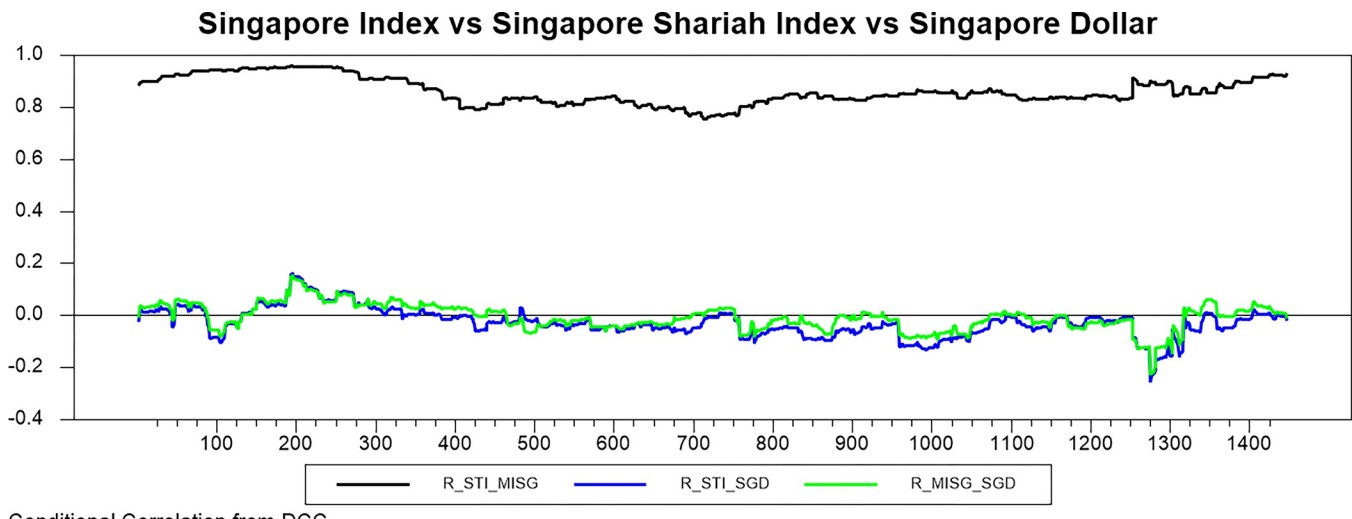

**Fig 11. Conditional correlation.**

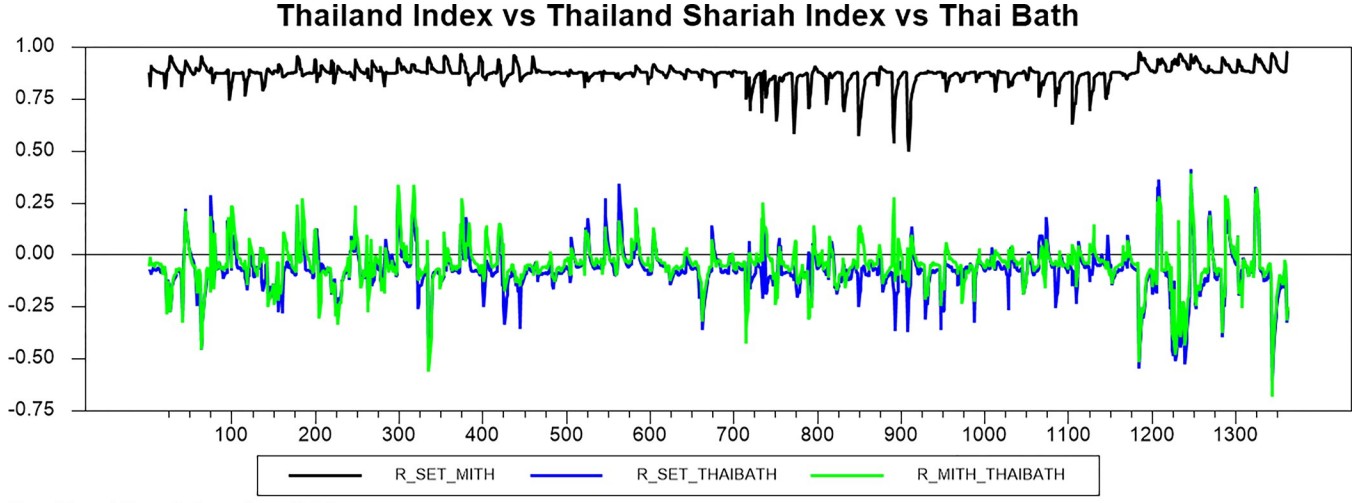

**Fig 12. Conditional correlation.**

## Hedge ratios and portfolio weights

After investigating the spillover and the comovements of three markets, in this section, we further examine how portfolio diversification is conducted using hedge ratios and portfolio weights. Kroner et al. [25] stated that the estimation of conditional covariance is used to develop the hedge ratios and optimal portfolio weights. To minimize the risk of a portfolio, a long position in one asset is hedged with a short position in a second asset. The risk-minimizing hedge ratio between two assets is as follows [25]:

$$\beta_{ij,t} = h_{ij,t}/h_{jj,t} \qquad [9]$$

Where $h_{ij,t}$ is the conditional covariance of i and j assets and $h_{jj,t}$ is the conditional variance of j asset at time t.

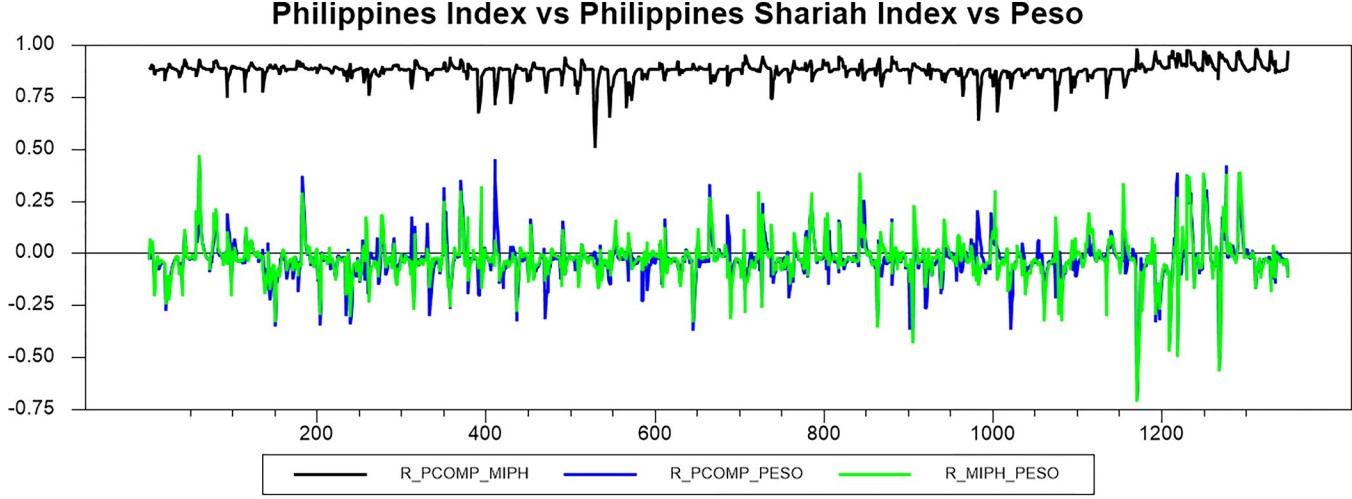

**Fig 13. Conditional correlation.**

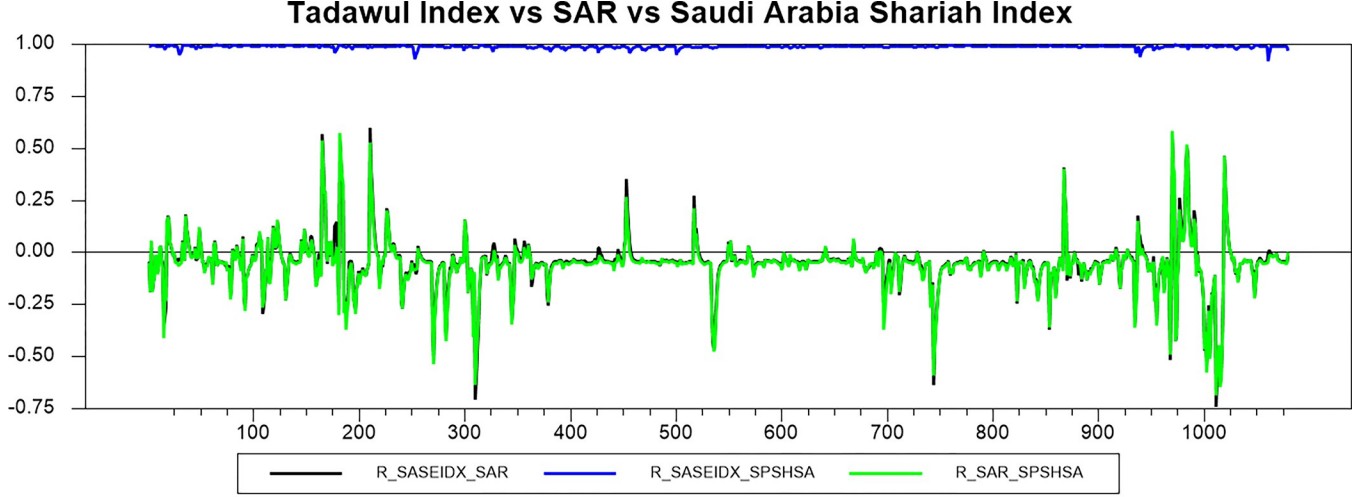

**Fig 14. Conditional correlation.**

The optimal portfolio weights between assets i and j are estimated as flows [25]

$$w_{ij,t} = (hh_{jj,t} - h_{ij,t})/(h_{ii,t} - 2h_{ij,t} + h_{jj,t}) \qquad [10]$$

with

$$w_{ij,t} = \begin{cases} 0, & \text{if } w_{ij,t} < 0 \\ w_{ij,t}, & \text{if } 0 \leq w_{ij,t} \leq 1 \\ 1, & \text{if } w_{ij} > 1 \end{cases} \qquad [11]$$

Where $w_{ij,t}$ is the optimal portfolio weight of the first asset and the optimal portfolio weight of the second asset, $1 - w_{ij,t}$. Tables 7 and 8 reported the hedge ratios and portfolio weights between currency, conventional and Shariah stocks computed based on the DCC model conditional variance above.

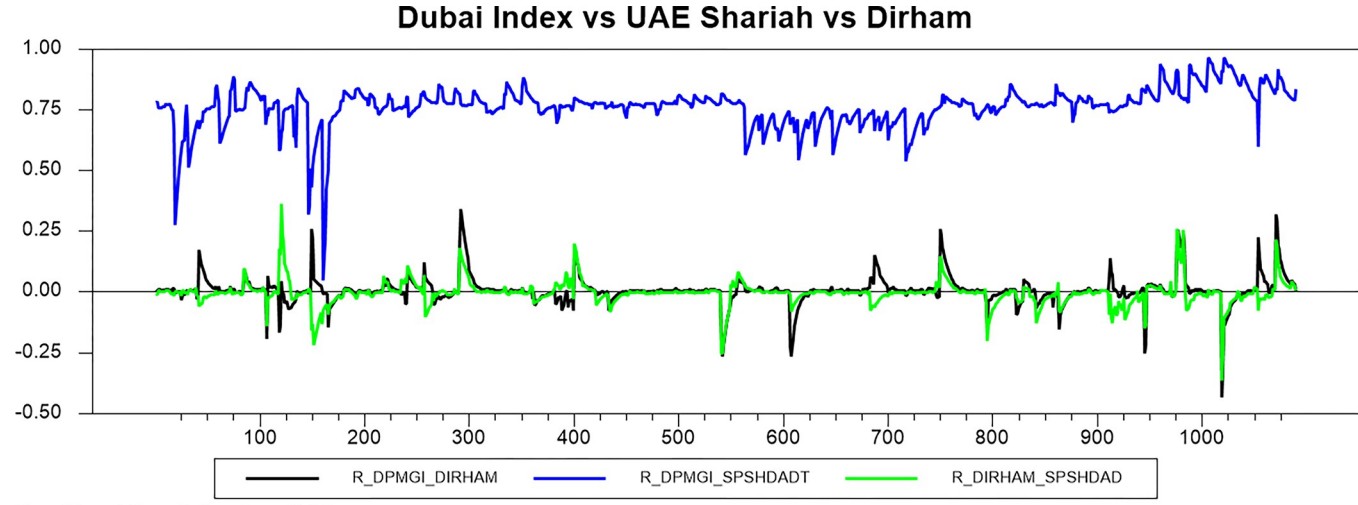

**Fig 15. Conditional correlation.**

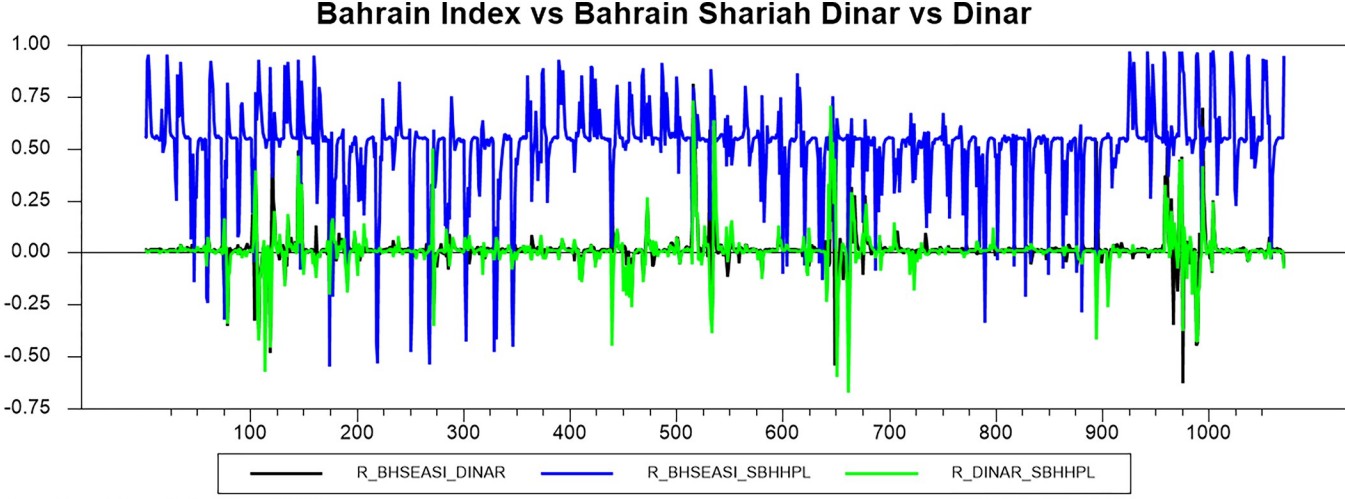

**Bahrain Index vs Bahrain Shariah Dinar vs Dinar**

Conditional Correlation from DCC

**Fig 16. Conditional correlation.**

Take one example of the hedge ratio from Table 7 for the Indonesian market. The hedge ratio value between conventional and Shariah stock is 0.72. It means that the investor would short Rp.72 in Shariah stock to hedge against a long position of Rp.100 in the conventional stock. Furthermore, the sign of the hedging ratio shows the position in the investor portfolio. A negative hedging ratio means that the investor should take the opposite position. For instance, the hedge ratio between currency and conventional stock in the Indonesian market is -0.16; it suggests that the investor would short Rp 62 in currency to hedge against a long position of Rp 100 in the conventional stock. A zero (0) value of hedge ratio in pairs of currency and conventional stocks and currency and shariah stock in GCC countries explains that the investor should not position in both conventional and Shariah stock to hedge against a long position in currency. It is consistent with the corresponding value of DCC, i.e., the comovement between currency and both conventional and Shariah stock around 0. As we know, the GCC countries embrace the pegged currency regimes.

On the other hand, the investor should short the currency position in much higher value–more than triple–to hedge against a long position in both conventional and Shariah stocks. In the rest of the countries (ASEAN-5 countries), Indonesia posits the highest hedge ratios for all a pair of assets, except a hedge ratio between conventional and Shariah stocks; Thai has a slightly higher value. Overall, the hedge ratios analysis suggests that, on average, both pairs of currency/conventional stock and currency/shariah stock assets are less costly to offset potential losses.

**Table 7. Hedge ratios.**

| Instrument | Indonesia | Malaysia | Singapore | Thailand | Philippine | Saudi Arabia | UAE | Bahrain |
|---|---|---|---|---|---|---|---|---|
| | Mean | Mean | Mean | Mean | Mean | Mean | Mean | Mean |
| Currency/Conv. stock | -0.16 | -0.01 | -0.01 | -0.01 | -0.002 | -0.00 | 0.00 | 0.00 |
| Conv. stock/currency | -1.28 | 0.89 | -0.34 | -0.97 | -0.53 | -31.75 | 43.28 | 4.37 |
| Currency/shariah stock | -0.12 | -0.01 | -0.01 | -0.002 | -0.001 | -0.00 | -0.00 | 0.00 |
| Shariah stock/currency | -1.66 | -0.56 | -0.52 | -0.96 | -0.54 | -33.96 | -16.39 | 6.19 |
| Conv. stock/shariah stock | 0.72 | 0.79 | 0.71 | 0.55 | 0.81 | 1.025 | 1.09 | 0.24 |
| Shariah stock/conv.stock | 1.23 | 0.89 | 0.89 | 1.39 | 0.98 | 0.96 | 0.56 | 1.16 |

**Table 8. Portfolio weights.**

| Instrument | Indonesia | Malaysia | Singapore | Thailand | Philippine | Saudi Arabia | UAE | Bahrain |
|---|---|---|---|---|---|---|---|---|
| | Mean | Mean | Mean | Mean | Mean | Mean | Mean | Mean |
| Currency/Conv. stock | 0.79 | 0.96 | 0.96 | 0.98 | 0.99 | 0.99 | 0.99 | 0.99 |
| Currency/shariah stock | 0.84 | 0.96 | 0.96 | 0.99 | 0.99 | 0.99 | 0.99 | 0.99 |
| Conv. stock/shariah stock | 1 | 0.57 | 0.72 | 0.98 | 0.83 | 0.07 | 0.1 | 0.97 |

Table 8 shows that portfolio weights for both currency/conventional stock and currency/shariah stock portfolios are in the range of 0.79–0.99. For instance, the Indonesian market's average weight for currency/conventional stock is 0.79, indicating that for a 100% portfolio, 79% should be invested in rupiah and 21% in conventional stock. The outcome of portfolio holding is consistent with Chkili's [13] suggestion for optimal portfolio holding with having more currency than stock. By combining this proportion, the investors will minimize portfolio risk without sacrificing the return expectation.

Moreover, the range of portfolio weights for conventional/shariah stock is wider, from 0.07 to 1. Indonesia's market poses the highest portfolio weight proportion, i.e., as we noticed, the correlation of movement between conventional and Shariah stock is almost 1. Hence, it is unsurprising that the investor should invest 100% in conventional or Shariah stock for the Indonesian market.

## Conclusion

International equity investments have been rapidly growing in the last three decades. It leads to an increase in the interdependency between stock returns and currency returns. In the meantime, the development of the Shariah stock market has also been growing swiftly. Hence, the interdependence among the three markets is highly possible. Significantly, the behaviour of the Shariah stock market investors is not significantly different from the conventional market. The increasing interdependency brings the consequences of volatility transmission among those markets. In this study, we have examined the comovement and spillover of volatility between the conventional Sharia stock indices and currencies, seeking to provide insights into the interplay between these critical components of the financial landscape. Our analysis has unveiled several noteworthy findings.

First, the results show that both Shariah and the conventional stock market respond the same way to the foreign exchange markets in each country; that is, the conditional variance of the currency market and the conventional stock market depends on their past volatility. A bidirectional (two-way relationship) volatility spillover exists only in the Malaysian market. Meanwhile, a unidirectional (one-way relationship) transmission is observed in the Indonesia, Singapore, Thailand, and Bahrain markets. The rest of the markets–the Philippines, Saudi Arabia, and UAE–do not have any volatility spillover evidence. Regarding spillover between conventional and Shariah stock markets, three ASEAN countries, namely Indonesia, Malaysia, and Singapore, and one of the GCC countries–Saudi Arabia–exhibit the spread of volatility.

Based on the DCC outcome, the conventional and Shariah stock in ASEAN countries and GCC countries unveils the markets' efficiency; only Bahrain shows a less efficient market. It implies no advantage of portfolio diversification in these markets. Contrarily, currency and stock (conventional and Shariah) markets provide portfolio diversification benefits for all ASEAN and GCC countries. The hedge ratios analysis infers that the least cost to offset potential losses is in both pairs of currency/conventional stock and currency/shariah stock.

This study has practical implications for investors, risk managers, policymakers, and financial institutions. Investors can use these research insights to build more diverse portfolios. Investors can lower risk by owning conventional/shariah stock and currency assets. As for risk managers and financial institutions, the study helps them better manage risk. They can employ risk management strategies such as hedging or asset allocation adjustments.

Furthermore, the findings can be used by regulatory agencies and central banks to make better-informed policy choices. For instance, they can think about how their monetary policies can affect these dynamics if they notice that changes in foreign currency prices significantly impact conventional/Sharia stock indexes. The study insights also can guide decisions related to trade agreements, exchange rate management, and economic development strategies. Finally, the study can lead to improving existing models and creating new ones to explain the complex interactions between asset classes and financial markets.

## Acknowledgments

The author would like to thank Prince Sultan University for their support.

## Author Contributions

**Conceptualization:** Nevi Danila.

**Data curation:** Nevi Danila.

**Formal analysis:** Nevi Danila.

**Investigation:** Nevi Danila.

**Methodology:** Nevi Danila.

**Writing – original draft:** Nevi Danila.

**Writing – review & editing:** Nevi Danila.

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
