## [Decision Letter · Decision Letter 0]

18 Sep 2023

PONE-D-23-26499

Spillover of Volatility among Sharia Stock Indices and Foreign Exchange Rates: ASEAN and GCC Market Study

PLOS ONE

Dear Dr. Danila,

Thank you for submitting your manuscript to PLOS ONE. After careful consideration, we feel that it has merit but does not fully meet PLOS ONE’s publication criteria as it currently stands. Therefore, we invite you to submit a revised version of the manuscript that addresses the points raised during the review process.

We look forward to receiving your revised manuscript.

Kind regards,

Ricky Chee Jiun Chia

Academic Editor

PLOS ONE

Journal Requirements:

"The authors would like to acknowledge the support of Prince Sultan University for paying the Article Processing Charges (APC) of this publication and for their support."

"The author(s) received no specific funding for this work"

4. Please ensure that you include a title page within your main document. You should list all authors and all affiliations as per our author instructions and clearly indicate the corresponding author.

5. Please amend your manuscript to include your abstract after the title page.

6. Please include a separate caption for each figure in your manuscript.

Reviewers' comments:

Reviewer's Responses to Questions

**Comments to the Author**

1. Is the manuscript technically sound, and do the data support the conclusions?

Reviewer #1: Yes

Reviewer #2: Yes

2. Has the statistical analysis been performed appropriately and rigorously? 

Reviewer #1: No

Reviewer #2: Yes

3. Have the authors made all data underlying the findings in their manuscript fully available?

Reviewer #1: Yes

Reviewer #2: Yes

4. Is the manuscript presented in an intelligible fashion and written in standard English?

Reviewer #1: Yes

Reviewer #2: No

5. Review Comments to the Author

Reviewer #1: Dear Author(s)

I would like to thank the author(s) for your submission and appreciate the opportunity to read and review your manuscript. I enjoyed reading it. From my point of view, it is a very interesting topic. However, I find some recommendations: my comments concern only with clarity, text, and interpretation of results. I would recommend publication, after the issues below are addressed.

Major clarifications:

1. The author(s) are highly advised to mention the implications for research, in the last section. I noted that the authors did not mention how this paper bridges the gap between theory and practice.

2. Since this paper is solving one of the issues, it is worthy of mentioning how can the research be used in practice? Are there implications consistent with the findings and conclusions of the paper?

3. The research results need more improvement to present the study findings. Therefore, more effort should be invested in the empirical findings and conclusion sections and tie together the other elements of the paper.

More comments can raise below:

1. The title: a good title should be concise, accurate, and informative, and it should tell the reader exactly what the article is about. Dear authors please rewrite the title.

2. Abstract: it does not provide an accessible summary of the study.

3. The Keywords: the keywords does not accurately reflect the content of the study.

Dear author(s) please try to rewrite the title, abstract and keywords to be more optimized for the study purposes.

4. Dear author(s) to help guide your readers, please end your introduction with the structure of the paper.

5. Data and Methods section: it needs more clarifications.

6. Empirical Findings and Conclusion sections: should be better analyzed and developed further, it needs more improvement to present the study findings. Please augment its quality with more depth, rigor and substance.

7. Please be careful in using the abbreviations and abbreviated words throughout the paper.

8. Please be careful in using "Punctuating", e.g. (comma, full stop, etc…) throughout the paper.

Finally, based on the above-mentioned comments, I suggest “major revision” of the paper before publication.

Reviewer #2: - I suggest not using abbreviations in the abstract, like GCC and ASEAN. You can write the name in full and in parathesis the abbreviation. Is custom in a paper to mention the whole name + in parenthesis the abbreviation, and then you can use the abbreviation below in the paper. Also, in the abstract you repeat very annoying market. Please rephrase somehow.

- Proofread with an expert the paper, because you have mistakes like ‘’Studies also reports’’ (Studies report), or phrases that are hard to be understood.

- Add a phrase in the Introduction why do you think your study improves the literature, what did you do differently than other authors. It is not clear in your introduction, as you used some concepts presented also by Hasan (2019)!

- Cut ‘’Here are;; …. In the introduction at the beginning of the third paragraph. It doesn’t look to academic.

- Before section Literature Review you say ‘’ hedge ratios and the optimal portfolio weight are estimated to convey the implication’’, but in the Data and Methods and Findings you mentioned weights. Why is it here in the singular?

- In the literature review you use ‘’Two theories linked the stock prices and the currency is the flow-oriented and the stock-oriented model.’’ Is confusing, because there is a difference between theory and model. Also ‘’On the other hand, Japan finds no evidence of volatility transmission.’’ Japan is not a person.

- In the data and methods section mention why did you not use all the ASEAN or GCC members? An explanation why did you remove some of the countries has to be presented!

- I suggest to you to use the same abbreviation everywhere for GARCH-BEKK or BEKK-GARCH. The same for GARCH-DCC, because you have it like DCC-GARCH. Use the correct form.

- Be clearer when you mention ‘’ Moreover, the information transforms from currency’’. ‘’Transforms’’ is quite ambiguous.

- ‘’their investment thru diversification’’ – through is more academic.

In general, the subject of the paper is interesting, but more emphasis should be made for policymakers. Good luck with your revision.

6. PLOS authors have the option to publish the peer review history of their article (what does this mean?). If published, this will include your full peer review and any attached files.

Reviewer #1: No

Reviewer #2: **Yes: **Florin-Teodor Boldeanu

---

## [Author Response · Author response to Decision Letter 0]

30 Sep 2023

Suggestion from Reviewer 1

1. The author(s) are highly advised to mention the implications for research, in the last section. I noted that the authors did not mention how this paper bridges the gap between theory and practice: Done – see the tracked changes in the conclusion

2. Since this paper is solving one of the issues, it is worthy of mentioning how can the research be used in practice? Are there implications consistent with the findings and conclusions of the paper?: Done - refer to the previous response

3. The research results need more improvement to present the study findings. Therefore, more effort should be invested in the empirical findings and conclusion sections and tie together the other elements of the paper: Done – see the tracked changes in the results and discussion section 

4. The title: a good title should be concise, accurate, and informative, and it should tell the reader exactly what the article is about. Dear authors please rewrite the title: Done- see the changes in the title 

5. it does not provide an accessible summary of the study: The abstract is reflected the summary of the study

6. The Keywords: the keywords do not accurately reflect the content of the study: Done – see the changes in keywords

7. Dear author(s) to help guide your readers, please end your introduction with the structure of the paper: Done – see the changes in the last paragraph of introduction

8. Data and Methods section: it needs more clarifications: Data – the explanation of choosing three GCC countries is added (see note under table 1). As for ASEAN, we use ASEAN-5 as sample of the study.

9. Empirical Findings and Conclusion sections- should be better analyzed and developed further, it needs more improvement to present the study findings. Please augment its quality with more depth, rigor and substance: Done – see the changes in results and discussion, and the conclusion sections

10. Please be careful in using the abbreviations and abbreviated words throughout the paper: Done – see the changes throughout the article

11. Please be careful in using "Punctuating", e.g. (comma, full stop, etc…) throughout the paper: Done 

Suggestion from Reviewer 2 

1. I suggest not using abbreviations in the abstract, like GCC and ASEAN. You can write the name in full and in parathesis the abbreviation. Is custom in a paper to mention the whole name + in parenthesis the abbreviation, and then you can use the abbreviation below in the paper. Also, in the abstract you repeat very annoying market. Please rephrase somehow: Done – see the changes in the abstract and throughout the paper

2. Proofread with an expert the paper, because you have mistaken like ‘’Studies also reports’’ (Studies report), or phrases that are hard to be understood: Done 

3. Add a phrase in the Introduction why do you think your study improves the literature, what did you do differently than other authors. It is not clear in your introduction, as you used some concepts presented also by Hasan (2019)!: The research gap is elaborated in the second paragraph, starts from second sentence. 

4. Cut ‘’Here are; …. In the introduction at the beginning of the third paragraph. It doesn’t look to academic: Done – see the changes in the beginning of third paragraph

5. Before section Literature Review you say ‘’ hedge ratios and the optimal portfolio weight are estimated to convey the implication’’, but in the Data and Methods and Findings you mentioned weights. Why is it here in the singular?: Done – adjusted accordingly

6. In the literature review you use ‘’Two theories linked the stock prices and the currency is the flow-oriented and the stock-oriented model.’’ Is confusing, because there is a difference between theory and model. Also ‘’On the other hand, Japan finds no evidence of volatility transmission.’’ Japan is not a person: Done – adjusted accordingly, see the changes in the literature review

7. In the data and methods section mention why did you not use all the ASEAN or GCC members? An explanation why did you remove some of the countries has to be presented!: Done – ASEAN refers to ASEAN-5 and the reason for selecting 3 GCC countries is elaborated under table 1 (see the note). 

8. I suggest to you to use the same abbreviation everywhere for GARCH-BEKK or BEKK-GARCH. The same for GARCH-DCC, because you have it like DCC-GARCH. Use the correct form: Done – see the changes throughout the paper

9. Be clearer when you mention ‘’ Moreover, the information transforms from currency’’. ‘’Transforms’’ is quite ambiguous: Done – change to “converted”, see pg. 8, first paragraph 

10. ‘’their investment thru diversification’’ – through is more academic.: Done – change accordingly, see the last paragraph in conclusion 

11. In general, the subject of the paper is interesting, but more emphasis should be made for policymakers: Done – see the last paragraph in conclusion

---

## [Editor Report · Decision Letter 1]

3 Oct 2023

Spillover of Volatility among Financial Instruments: ASEAN-5 and GCC Market Study

PONE-D-23-26499R1

Dear Dr. Nevi Danila,

We’re pleased to inform you that your manuscript has been judged scientifically suitable for publication and will be formally accepted for publication once it meets all outstanding technical requirements.

Kind regards,

Ricky Chee Jiun Chia

Academic Editor

PLOS ONE
---

## [Editor Report · Acceptance letter]

11 Oct 2023

PONE-D-23-26499R1 

Spillover of Volatility among Financial Instruments: ASEAN-5 and GCC Market Study 

Dear Dr. Danila:

I'm pleased to inform you that your manuscript has been deemed suitable for publication in PLOS ONE. Congratulations! Your manuscript is now with our production department. 

Kind regards, 

on behalf of

Dr. Ricky Chee Jiun Chia 

Academic Editor

PLOS ONE